# Towards Increasing Residential Market Transparency: Mapping Local Housing Prices and Dynamics

**Radoslaw Cellmer [1] and Radoslaw Trojanek [2,*]**

[1]  Institute of Geospatial Engineering and Real Estate, University of Warmia and Mazury in Olsztyn, Prawocheńskiego 15, 10-720 Olsztyn, Poland; rcellmer@uwm.edu.pl

[2]  Institute of Economics, Department of Microeconomics, Poznań University of Economics and Business, Al. Niepodległości 10, 61-875 Poznań, Poland

*   Correspondence: r.trojanek@ue.poznan.pl

**Abstract:** This article attempts to use spatial maps as a way of presenting additional information about the phenomena occurring in the housing market. In our opinion, spatial maps may facilitate understanding and provide more detailed information, which undoubtedly should increase the transparency of the housing market. The study used 12,219 transactions of apartments in Poznań in the years 2013–2017. General principles of price visualization activity and housing market dynamics were established in this study. The map of prices may reflect the location values determined by the quality of the urban infrastructure, distance from specific locations, and environmental factors. Market activity maps reveal areas where the market is dynamically developing, while information on trends in the number of transactions and price changes may demonstrate the growing or declining attractiveness of areas. The research is based on a model of hedonic regression in the form of ordinary least squares (OLS), quantile regression (QR), and geographically weighted regression (GWR). The maps presented should increase the transparency of the residential market (e.g., by providing more detailed information). However, one should bear in mind the limitations in the use of these methods resulting from a small number of transactions in a thin market.

**Keywords:** housing market; market transparency; price map

## 1. Introduction

The residential market in 2015 accounted for as much as 75% of the value of the real estate market and 44% of the value of the world's significant assets [1]. At the same time, the number of people owning houses has increased significantly in recent decades due to the liberalization of housing credit markets as well as social and political changes in individual countries [2].

The housing market is undoubtedly the most crucial segment of the real estate market, which is a result of the fact that, among other things, housing is a good that stands out from other goods produced and consumed by humans. It satisfies a basic need, which is shelter. It should be noted that a flat is a highly capital-intensive good, which creates problems at the stage of satisfying this need; hence, satisfying the whole complex of needs related to it constitutes an important place in the budgets of households. Considering the above, as well as the imperfection of the housing market, investment decisions of households are made in conditions of lack of access to full information.

The main reasons for the low efficiency of the real estate market are primarily [3]: the local nature of the real estate market, the relatively small number of transactions concluded on this market, significant discrepancies in the expectations of market participants, the vast role of financial institutions, and the uniqueness and lack of comparability of investments in this market. Moreover, as Linneman [4] points out, the lack of experience of buyers and sellers, significant differences in the technical condition

of buildings, or non-investment motives for making investments may lead to significant disproportions in information on the real estate market. The effective market hypothesis refers to the category of the perfect market, which is closely related to the conditions of free competition.

Real estate market imperfection undoubtedly stems from the uniqueness of real estate as a commodity and in particular by the physical characteristic of the inability to transfer real estate in space and the associated rigid supply. The imperfection of the real estate market is also determined by the fact that transactions on this market require large financial capital, including external funding. In this situation, interest rates and availability of loans have a significant impact on the price. There are often unreasonable behaviors of buyers and sellers, who do not make their decisions solely on the basis of price; it is often fashion, neighborhood, tradition, or advertising that decides. The lack of transparency of this market is also influenced by a large number of private transactions, the course of which usually remains undisclosed. The limited information transparency of the residential market means that most buyers and sellers do not have sufficient data on property prices at the time of buying and selling decisions, as obtaining it is labor-intensive and requires professional experience. Moreover, the real estate market is an area of widespread public intervention [5].

The concept of transparency is often unclear and has different meanings, and various aspects of transparency can be found in different areas [6]. According to Schulte et al. [7] (p. 91): "Real estate markets can be described as transparent when it becomes clear how the market mechanisms and the variables behind these mechanisms work, i.e., when there is as much information as possible available at any point in time." In terms of the real estate market, this leads to the necessity of obtaining access to reliable data related to the following sub-markets: the rental market, the investment market, the property and construction market, and the capital market.

The transparency of real estate has been increasing both in the local as well as in global markets. In general, strong evidence of the improved transparency of European real estate markets is demonstrated by the fact that some of the European developing markets are among the top improvers [8,9].

Lindqvist [6] recognizes five aspects of transparency in residential property transactions: transparency in transaction procedures, including access to information and assistance in the process; transparency in legal information; transparency in financing; transparency in taxation; and transparency in transaction costs.

Recent technological advances and improved data collection methods have brought many developments from an applied and analytical viewpoint to the analysis of real estate [10]. Vast volumes of data are gathered, transformed, and analyzed to forecast market trends.

Moreover, access to public resources has become easier. In many countries, real property data has significantly improved in terms of geographic coverage, depth, granularity, availability of new information, currency of information, and the overall integrity and accuracy of available data sources [11]. In the case of Poland, there was a massive change in the possibility of obtaining information on property transactions. Nowadays, these data are easy to access in electronic form from the Land Registry, while 10 years ago it was necessary to rewrite property transaction information directly from notarial deeds. This information has limitations in the description of properties; however, it is still a huge step forward. Taking into account the ease of gathering data from different online sources (e.g., information on properties or sales, GIS (geographic information system) data-coordinates, layers shared by municipalities, or data from the OpenStreetMap project) there is an increased possibility of merging data, estimating models with new variables, and elaborating new ideas and solutions [12,13].

Taking into account the above considerations, the paper attempts to use maps to improve the availability of information about the housing market and thus increase the transparency of this market. The developed maps should increase the residential market transparency as they provide new information to potential buyers, owners, investors, and in this way, reduce the investment risk in this market. Households making investment decisions are influenced by budget constraints and preferences. Currently, access to information on the housing market is much better than it was a

dozen or so years ago. It must be stated that access to information on housing price trends has been improved as, since 2013, European Union member states, under Regulation (EC) No. 93/2013, have been obliged to publish housing price indexes [14,15]. These indexes provide information about apartment price changes in cites. The complexity of housing markets, legal issues, and construction of new infrastructure all leads to different development of apartment prices in the market. Thanks to the maps developed, we are able to follow these changes in a spatial context.

## 2. Literature Review

Open access to information on the prices and values of properties is one of the underlying conditions for ensuring the transparency of the property market. The critical role of monitoring real estate markets and making market information available to the public is emphasized by Gaffney [16], pointing to the relationship between local decisions concerning space and national and even global economic consequences. Access to spatial information on prices and values allows one, to a large extent, to control the risk of investing in real estate, especially in a dynamic economic situation [17].

Real estate information provided in the form of maps can be particularly useful for public administrations, but also real estate appraisal agencies, mortgage lenders, and planning organizations [18]. Real estate, as well as their prices, are linked by topological relations, which allow relating information on the price of real estate to geographical space and its visualization in the form of a map with the use of GIS tools. Price and value maps are most often discussed in the context of mass valuation of real estate for tax purposes [19]. While the development of maps for tax purposes requires the application of rules strictly defined by law, investment or generally utilitarian purposes open up the possibility of applying open rules based on substantive analysis of local real estate markets. The basis for the development of such maps is usually detailed market analyses prepared for areas of single cities [20].

In the mainstream research on cartographic visualization of prices and values, hedonic models are used, taking into account selected characteristics of real estate as price determinants [21,22]. It is then assumed that real estate is a heterogeneous good, the price of which is decomposed into a set of utility features constituting the explained variables [23,24].

Prices and values of real estate depend on many exogenous and endogenous elements, which should be taken into account in the market analysis, with a particular emphasis on detailed location [25–28]. One of the essential elements of such an analysis is to isolate the impact of location factors that constitute the essence of cartographic reflection of value [29]. Considerations on location in the context of determining the determinants of apartment prices are presented, among others, by Kiel and Zabel [30] and Kolbe et al. [31], indicating at the same time relatively simple methods of econometric modelling as a primary tool for identifying location factors. The underlying assumptions in the methodology for preparing price and property value maps may, therefore, be based on the knowledge of the relationship between prices and location values resulting, among others, from the distance from particular areas [32]. These distances can be expressed as a geometric feature or, respectively, as travel time [33]. The basis for developing a value map can then be zones determined on the basis of distances from, among others, the city center, main streets, areas influencing the value, as well as the location of areas threatened by floods. Liu et al. [20] assess the development of prices and values depending on the distance from the CBD (central business district), elements of social infrastructure, schools, etc. Jim and Chen [34] emphasize the key importance of environmental elements including distance to green areas, water, and noise exposure. The role of the environment, especially lake areas in urban areas, is also highlighted by Zhang et al. (2015). Taking these factors into account in the market analysis allows for the valorization of urban space, which at the same time reflects the value of real estate.

Against the background of this trend of research, it seems particularly interesting to use GIS tools and geostatistical methods to model surfaces showing the value of real estate. Hedonic models are influenced by limited data due to high monetary and time data collection costs, geostatistical models,

explicit modelling of spatial autocorrelational effects, as well as spatially extensive patterns, which are compelling reasons to develop an alternative. In contrast, geostatistical approaches (co-kriging) allow for the assessment of auxiliary variables that are spatially codependent on property values, including structural and residential housing characteristics. Bourassa et al. [25] and Tsutsumi et al. [32] advocate for the combined use of hedonic models and geostatistical methods. Geostatistical methods can be treated as a natural complement to traditional statistical analysis, taking into account the spatial distribution of the analyzed phenomenon. These methods are much less commonly used in the real estate market than other statistical methods [18]. A specific obstacle to their use may be the need to meet fundamental assumptions concerning the size of the data set and, above all, the location [35].

The spatial structure of location features may also be included in the geographically weighted regression (GWR) model [36,37]. Due to price volatility over time, Fotheringham et al. [38] postulate the use of GWR not only for spatial analyses but also for spatial and temporal analyses. These analyses allow us to present not only the current spatial distribution of prices and values but also local trends and short-term forecasts.

It should be stressed that no universal methods for mapping land prices and values have been developed so far, nor have dynamic maps been developed to take account of the specificities of the real estate market. The main problems that arise during the preparation of such maps include the lack of sufficient reliable data from local real estate markets, the uneven spatial distribution of the properties being traded, or imperfect methods and tools [39].

## 3. Methodology

The research is based on a model of hedonic regression. Hedonic regression's first recorded use dates back to 1922, when the farmland price model was created by G. A. Hass [40]. The first researcher to use the hedonic approach to evaluate the real estate market was likely Ridker, who aimed at identifying the effect of reducing pollution on house prices [41]. The methodological basis for the hedonic approach was established by Lancaster [23] and Rosen [24].

In this research, general principles of price visualization, activity, and dynamics of the real estate market were developed. To visualize the price level on the residential market in the form of a price map, regression modelling based on the QR (quantile regression) model was used, which was compared with the classical OLS (ordinary least squares) regression model. Based on the QR model, a map of predicted prices and a map of residuals were prepared as a basis for the final price map. The map of market activity was prepared using kernel density estimation, while the model of GWR (geographically weighted regression) was used to develop the map of price dynamics.

Transaction prices, in general, are affected by both non-spatial features and the location values of the property, which can generally be described as follows:

$$P = P_{base} + P_{local} \tag{1}$$

where $P_{base}$ is the price of the model property excluding location factors (base price), and $P_{local}$ is the impact of location.

A model with a general form can, therefore, be expressed as follows:

$$Y = \beta_0 + \sum_{i=1}^{k} \beta_k X_k + \sum_{j=1}^{l} \beta_l X_l + \varepsilon \tag{2}$$

In this way, the base price can be determined directly based on the statistical model, while the analysis of the location impact should take into account the parameters of the model and a random component $\varepsilon$. The spatial distribution of a random component may be determined on the basis of spatial interpolation using the ordinary kriging method. It can be noted that the essential elements of data analysis can be integrated into a single model, which takes into account the interdependence between the results

of statistical modelling and estimation by kriging methods. The regression–kriging model, used to estimate prices for the purposes of map compilation, can in its general form be presented as follows, taking into account the logarithmic transformation:

$$Price = \exp\left(\beta_0 + \sum_{i=1}^{k}\beta_k X_k + \sum_{j=1}^{l}\beta_l X_l + \sum_{m=1}^{n} w_m(s_0)\varepsilon(s_m)\right) \tag{3}$$

$$Price = \exp(P_{base})\exp\left(\sum_{j=1}^{l}\beta_l X_l\right)\exp\left(\sum_{m=1}^{n} w_m(s_0)\varepsilon(s_m)\right) \tag{4}$$

where the expression $\sum_{m=1}^{n} w_m(s_0)\varepsilon(s_m)$ denotes the value of the residuals at $s_0$ based on the residuals at $s_m$. The price map, developed in the GIS environment, is in fact the product of the *exp(P_{base})* constant and two layers, the first of which is the result of statistical modelling and the second one is the result of geostatistical modelling.

### 3.1. Quantile Regression

The quantile regression proposed by Koenker and Bassett [42] allows estimation of various quantile functions of the conditional variable distribution. At any point in the distribution of the dependent variable, the determinants of the dependent variable can be defined [43]. The specific case of quantile regression for quantiles of 0.5 (medians) is equivalent to the least absolute deviation (LAD) estimator. In the case of heteroskedasticity, quantile regression estimation of 0.5 may be more effective than the OLS estimator [44]. The quantile regression technique is close to ordinary regression. The distinction lies in the way in which the margin of sums of squared residuals is checked; the quantile regression aims for the margin of the weighted sums of absolute residuals. The quantile regression is preferable to the alternative strategy, because heteroscedasticity, outliers, and unobserved heterogeneity can be handled [45].

### 3.2. Kernel Density Estimation

The results of spatial analysis of market processes, which may have a certain intensity in space, are crucial information in order to understand the conditions of local real estate markets. They can be presented in a cartographic form presenting, among others, the activity of the market expressed in the form of the number of transactions. One of the fundamental problems in estimating the intensity (or density) of a given phenomenon in space is the fact that these phenomena often have a point character, or their identification is possible only at selected measurement points. In order to determine the density, kernel estimation can be used, which allows one to explicitly take into account the spatial resolution. This estimation aims to model a smooth surface representing a density that is dependent on the concentration of points in the surrounding area [46]. The kernel probability density estimator in its basic form is defined by the following formula [47]:

$$\hat{f}(x) = \frac{1}{mh^n}\sum_{i=1}^{m} K\left(\frac{x - x_i}{h}\right) \tag{5}$$

where m is the random sample size, h is the smoothing parameter, and K is the function fulfilling the following conditions:

$$\int_{IR^n} K(x)dx = 1, K(x) = K(-x)\forall\, x \in IR^n, K(0) \geq K(x) \tag{6}$$

In the estimation of the density of a phenomenon, each measurement object is replaced by a value calculated according to the probability density function, and then function values are added to obtain an aggregate surface or continuous density field [48]. Depending on the purpose of the study, the type of data, and the expected results different forms of kernel functions are used, e.g., Epanechnikov kernel, monoblock, bivalve, or normal [49]. The value of the smoothing parameter has a fundamental influence on the quality of the kernel estimator. If the value is too low, a significant number of local extremes appears, which may contradict the actual properties of the real population. On the other hand, too high values of the h parameter result in excessive smoothing of the estimator, masking the specific features of the studied distribution [49]. Kernel estimation has been used for many years, especially in system analysis [50], but there are few studies presenting its application to spatial analyses of the real estate market. Examples of applications of this method include, for example, estimating population density and structure [51], access to retail centers [46], or analysis of access to public health services [52]. Modelling the surface area representing the density of spatial phenomena also allows determination of the correlation between these phenomena. An example is the analysis of relations between the spatial configuration of a communication system and various socio-economic indicators concerning the local community [53]. The use of kernel estimation for spatial analysis of the real estate market may consist not only in the assessment of density but also in the assessment of the intensity of a given phenomenon. This methodology may be especially useful for the creation of cartographic studies concerning, among others, the number of transactions, traffic intensity, the intensity of buildings, or changes in ownership in spatial terms [54].

*3.3. Geographically Weighted Regression (GWR)*

In traditional model regression for real estate analyses, possible relationships (spatial autocorrelation) that may occur between levels of the particular phenomenon in space are usually not explicitly taken into consideration and it is assumed that the stability of the mechanism refers to price formation in geographical space. The significance of traditional regression model parameters, in this situation, does not rely upon the spatial structure of the phenomena examined, which can lead to the findings being misinterpreted [55], in general with the presupposition of the spatial heterogeneity of property markets. One way to solve the problem of considering the spatial form of the phenomenon under study in regression models is to give weight to observations, which, due to their location in space, may theoretically have a greater influence on the examined phenomenon than others, which may be conveyed by the GWR. Geographically weighted regression is based on the premise that the parameters of the model can be measured independently at each point in space for which the values of the explanatory variables are known. The estimation of the model parameters at a given location is based on the assumption that the observations made at points located closer to the studied point will have a higher weight than the observations located further away [55]. The general equation of the GWR model is as follows:

$$Y = \beta_0(x_i, y_i) + \sum_{i=1}^{n} \beta_j(x_i, y_i) \cdot X_i + \varepsilon_i(x_i, y_i) \tag{7}$$

where $\beta_0(x_i, y_i)$ and $\beta_j(x_i, y_i)$ denote the parameters of the regression model at the coordinate points ($x_i$, $y_i$). GWR model parameters are estimated in a similar way as classical models, but location-dependent observation weights are taken into account, such that

$$\hat{\beta}(x_i, y_i) = (X^T W_{(i)} X)^{-1} X^T W_{(i)} Y \tag{8}$$

where $W_{(i)}$ is a weight matrix, which is a function of the distance between the location of a particular coordinate ($x_i$, $y_i$) and the location of each point of observation. The matrix shall take on a diagonal form in which the elements are the value of weights determined in such a way that they decrease

with the distance from the point at which the geographically weighted regression model is estimated. In most cases, functions similar in shape to Gauss's distributions are used to determine weights [55]:

$$w_{ij} = e^{-\frac{1}{2}\left(\frac{d_{ij}}{h}\right)^2} \text{ or } w_{ij} = \left[1 - \left(\frac{d_{ij}}{h}\right)^2\right]^2 \tag{9}$$

where $d_{ij}$ is the distance between the location $i$ and $j$, whereas $h$ denotes bandwidth. This parameter indicates the spatial range from which the observations will be accepted for calculation, which means that $w_{ij} = 0$ for $d_{ij} > h$.

## 4. Data

Poznań is situated in the Wielkopolskie Province in central–western Poland. It is the fifth-largest town in Poland by inhabitants (537,682 in 2018 year) and the eighth largest by size (262 sq km). Data on apartment sales from 2013 to 2017 were obtained from the Geodesy and Municipal Cadastre in Poznań. This research only refers to dwellings in multi-family buildings (apartments). In the case of Poland, the majority of dwellings are located in multi-family buildings (around 90% in the case of big cities). The scarcity of transactions, as well as lack of full descriptions of single-family houses in notarial deeds, could lead to biases and misinterpretation of the maps prepared using such a dataset. Data on transactions included all types of properties, both residential and non-residential (e.g., commercial properties or garages). In the data-cleaning process, sales of more than one residential unit and non-free market transactions (e.g., debt-collector sales) were removed. The final dataset contained 12,219 geocoded apartments transactions for the years 2013–2017 (Figure 1).

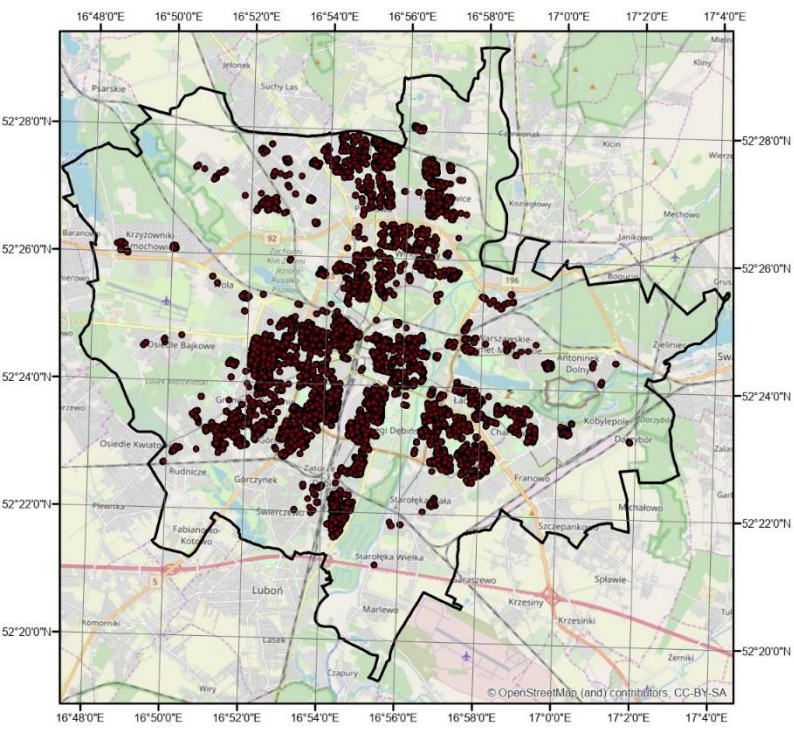

**Figure 1.** Apartment transactions in Poznań in 2013–2017.

The details included in notarial dwelling contracts contained the following items: the date of the sale, the price (in PLN, 1 EUR = 4.2359 PLN at the average exchange rate of the National Bank of Poland of 04.11.2019), the size of an apartment, and the information on auxiliary premises. Such a set of factors may bias the results of the research, as notarial contracts do not include information on strong pricing components such as construction technology. The addresses of the sales were geocoded using

the Google Maps API (application programming interface). Then, a lot of information on the height of buildings and the year of completion was applied based on the cadastral records. The Street View application on maps.google.com was used (virtual inspections of buildings) to supply the relevant data on height, completion year and, most importantly, construction technology. Additional analyses were carried out with the use of data from the Spatial Information System of Poznań city. Based on these data, distances from locations and objects (e.g., the center of the city, green areas, lakes, malls, etc.) were estimated. The list of variables accepted for analysis is presented in Table 1.

**Table 1.** Qualitative and quantitative variables applied in the models.

| Variable | Description |
|---|---|
| lnprice1m2 | Natural logarithm of price per 1 square meter of a dwelling |
| y2013, y2014, y2015, y2016, y2017 | 5-time dummy variables. If the apartment was sold in a given year, it takes the value 1; otherwise, it takes 0 |
| low, medium, high | 3 dummy variables. Low-intensity residential building (up to 4 floors), then low = 1, otherwise 0, High-intensity residential building (10 floors and more), then high = 1, otherwise 0. |
| age | Age of the building in years |
| area | Area of dwelling $m^2$ |
| tech | If the dwelling is located in a building made with prefabricated technology, it takes the value 0; otherwise, it takes 1. |
| dtownhall | Distance to the city hall in km |
| dgreen | Distance to urban green areas in km |
| dlake | Distance to lakes in km |
| dshops | Distance to a mall in km |
| LUA | The outside zone of Limited Use Area created around Pozań Ławica Airport |

In total, 14 explanatory variables were adopted (including the year of sale). The natural logarithm from the transaction price of 1 $m^2$ of usable area of the premises was used as an explained variable. Descriptive statistics of the adopted variables are presented in Table 2. Moreover, in Appendix A, the histograms of the chosen variables are provided.

**Table 2.** Descriptive statistics of variables applied in models.

| Variable | Min | Max | Mean | Std. Dev. |
|---|---|---|---|---|
| lnprice1m$^2$ | 6.660 | 9.528 | 8.531 | 0.256 |
| y2013 | 0.000 | 1.000 | 0.166 | 0.372 |
| y2014 | 0.000 | 1.000 | 0.170 | 0.376 |
| y2015 | 0.000 | 1.000 | 0.188 | 0.391 |
| y2016 | 0.000 | 1.000 | 0.249 | 0.432 |
| low | 0.000 | 1.000 | 0.656 | 0.475 |
| high | 0.000 | 1.000 | 0.147 | 0.354 |
| age | 1.000 | 212.000 | 43.341 | 34.175 |
| area | 15.140 | 297.58 | 52.248 | 20.383 |
| tech | 0.000 | 1.000 | 0.712 | 0.453 |
| dtownhall | 0.061 | 8.925 | 3.323 | 1.568 |
| dgreen | 0.010 | 1.030 | 0.284 | 0.187 |
| dlake | 0.240 | 6.000 | 2.400 | 1.025 |
| dshops | 0.010 | 5.207 | 1.194 | 0.645 |
| LUA | 0.000 | 1.000 | 0.060 | 0.238 |

Among the quantitative variables characterizing residential units, there was a significant difference in the age of the building (over 200 years) and the usable floor area (over 280 $m^2$). The analysis of the correlation between the variables showed that the vast majority of explanatory variables were significantly correlated with the response variable.

The research included the development of a map of real estate prices, a map of market activity, and a map of the dynamics of price changes. R environment and ArcGIS v. 10.4.1 software from ESRI were used for analyses and visualization.

## 5. Results and Discussion

### 5.1. Price Modelling and Mapping

When preparing the price map of residential units, it should be taken into account that the real estate market is extremely complex, and transactions involve objects that differ not only in location but also in features that are not spatial in nature (age, area, construction technology, etc.). Applying spatial interpolation directly to property prices, in this case, would not bring the expected results due to the heterogeneity of the objects. According to the adopted assumptions, the price map should present first of all the spatial differentiation of prices under the influence of location values, not non-spatial attributes. The problem of price differentiation can be solved by adjusting transactional prices in such a way that they refer to a model property with some unambiguous, predetermined characteristics [35]. These adjustments were based on the estimation of parameters of the quantile regression model; however, in the case of spatial variables (e.g., distance from characteristic places), the impact on prices may be presented directly on the map in the form of values resulting directly from the model. In the proposed methodology, a unique role was played by the so-called reference layer, which presented the spatial distribution of the value of properties with strictly defined non-spatial features. The paper assumed that the reference layer would refer to a flat of 50 m$^2$, located in a building 10 years of age, with an average height (from 5 to 10 floors), built using traditional technology.

The results of statistical modelling (OLS and QR models) are presented in Table 3.

**Table 3.** Estimation results (dependent variable is a natural logarithm of price per square meter). OLS is ordinary least squares; QR is quantile regression.

| | OLS | | | QR | | |
| --- | --- | --- | --- | --- | --- | --- |
| | **coef** | **std. Error** | ***p*-Value** | **coef** | **std. Error** | ***p*-Value** |
| const | 9.025 | 0.011 | <0.001 | 9.037 | 0.011 | <0.001 |
| y2013 | −0.141 | 0.006 | <0.001 | −0.131 | 0.006 | <0.001 |
| y2014 | −0.101 | 0.006 | <0.001 | −0.091 | 0.006 | <0.001 |
| y2015 | −0.088 | 0.006 | <0.001 | −0.080 | 0.006 | <0.001 |
| y2016 | −0.043 | 0.005 | <0.001 | −0.040 | 0.005 | <0.001 |
| low | −0.036 | 0.005 | <0.001 | −0.026 | 0.005 | <0.001 |
| high | −0.069 | 0.008 | <0.001 | −0.071 | 0.008 | <0.001 |
| age | −0.003 | 0.000 | <0.001 | −0.003 | 0.000 | <0.001 |
| area | −0.004 | 0.000 | <0.001 | −0.004 | 0.000 | <0.001 |
| techn | 0.093 | 0.006 | <0.001 | 0.094 | 0.005 | <0.001 |
| dtownhall | −0.011 | 0.002 | <0.001 | −0.008 | 0.002 | <0.001 |
| dgreen | −0.019 | 0.010 | 0.059 | −0.011 | 0.011 | 0.227 |
| dlake | −0.022 | 0.002 | <0.001 | −0.019 | 0.002 | <0.001 |
| dshops | −0.017 | 0.003 | <0.001 | −0.026 | 0.003 | <0.001 |
| LUA | −0.008 | 0.008 | 0.322 | −0.014 | 0.008 | 0.049 |
| LogLik | | 2221.525 | | | 3189.830 | |
| AIC | | −4411.049 | | | −6349.660 | |

In both the OLS and QR models, almost all variables turned out to be statistically significant. The difference occurred only in the case of the limited use area (LUA) variable, which in the QR model was significant at p-values less than 0.05, while in the OLS model, it turned out to be statistically insignificant. Coefficients signs were justified, as well as the order of magnitude of parameters. The influence of particular variables was associated with expectations considering the physical features of the apartments.

In addition, the location in LUA, the increase in distance from the city center, urban green areas, and lakes negatively impacted the apartment's price. This supports the findings of earlier studies in Poland [56–63]. The differences in parameter values between OLS and QR models were small. It was decided to use the QR model in the further analysis mainly due to the way that it treated outlier observations, which are a common phenomenon in the real estate market, as well as the evaluation of the model according to the AIC information criterion. It should also be noted that when comparing these models, due to the method of estimation, the criterion of the coefficient of determination and minimization of errors (residuals) should not be used. The standard deviation of the residuals in both models was similar to 0.202, which is 2.4% of the average logarithm of the 1 m$^2$ price.

The model price layer was obtained by substituting in the regression model the values of features corresponding to the reference property and the values resulting from rasters representing distances. The residual layer was obtained by spatial interpolation using the ordinary kriging method. Figure 2 presents a map of prices (in natural logarithms) resulting from the model and a map of the spatial distribution of the residuals.

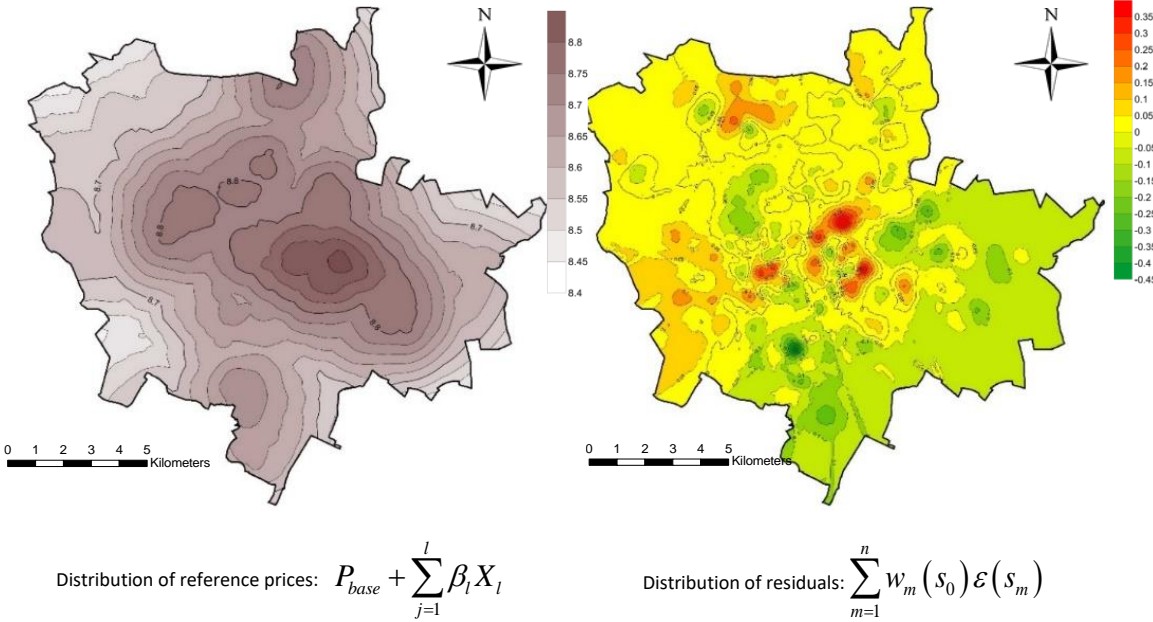

Distribution of reference prices: $P_{base} + \sum_{j=1}^{l} \beta_l X_l$　　　　Distribution of residuals: $\sum_{m=1}^{n} w_m(s_0)\varepsilon(s_m)$

**Figure 2.** Map of reference prices and map of residuals.

The results of the regression analysis suggest that the highest prices were found to be in the center and decreasing as we move towards the outskirts of the city. The price distribution was also significantly influenced by the city's spatial structure, including the distribution of green areas and lakes. The highest concentration of positive residues was recorded in the central parts of the city, which at the same time are perceived as the most attractive. Negative residuals concern mainly the areas in the southern and eastern parts of the city, which are related mainly to the difficulties in transport accessibility, as well as to the aircraft noise from the military airport Krzesiny [64].

The final price map was created by superimposing the reference price layers and the spatial distribution layers of the residuals obtained through spatial interpolation using the kriging method.

As supplementary information, a map of average prices in particular surveying districts was prepared, in which zone statistics of price maps were used. These maps are presented in Figure 3.

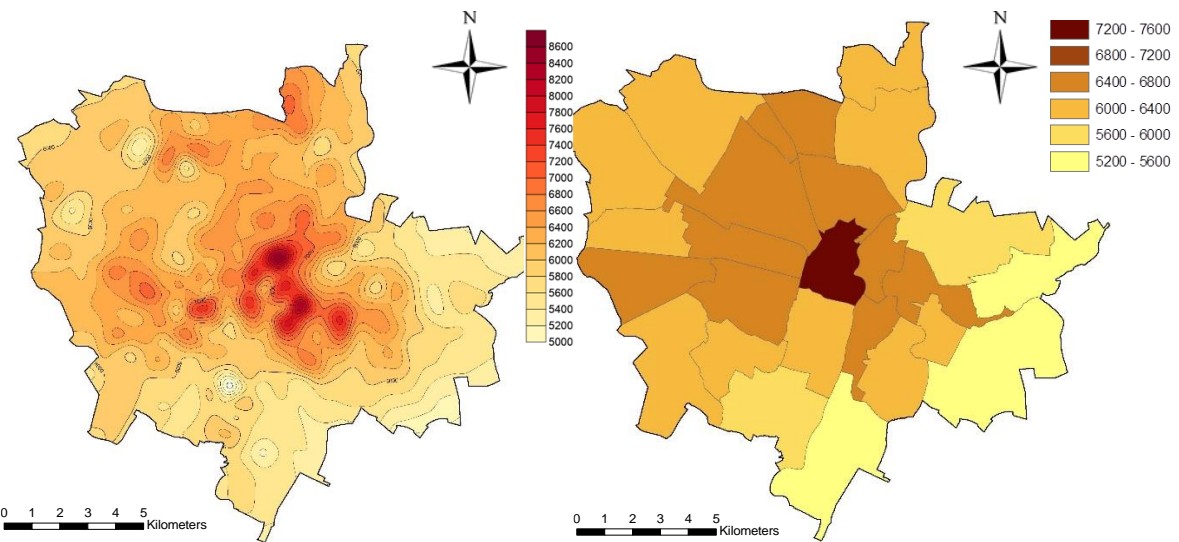

**Figure 3.** Price map of apartments. The values are given in PLN (Polish Zloty) per sq m.

## 5.2. Mapping Market Activity

Market activity was shown on maps as a distribution of the density of points corresponding to the location of the transaction. For this purpose, the kernel function in the form of a two-weighted kernel [47] was used. The smoothing parameter was adopted as h = 1000 m. As a result, a continuous surface was obtained, which represents the density of the analyzed phenomenon. Due to the fact that the analysis covered five years, the average density calculated independently for each year of the analysis was presented. As a supplement, a map of the average annual change in market activity, represented by the average annual increase (or decrease) in the number of transactions per unit area, expressed in absolute values, was prepared. These maps are presented in Figure 4.

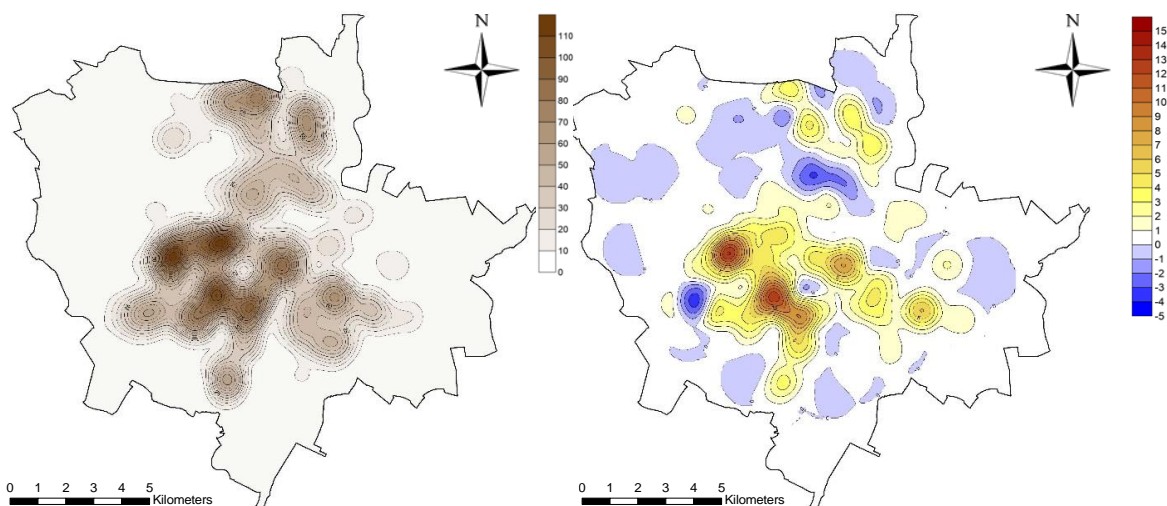

**Figure 4.** Market activity map. Left: average annual number of transactions. Right: average annual change in the number of transactions.

The areas with the highest market activity naturally reflect the potential supply dominated by high-intensity residential developments. The highest increase in the number of transactions per unit of area was recorded there. It is easy to see that the spatial differentiation of market activity is primarily influenced by the spatial structure of the existing stock. However, this diversity is caused not only by potential supply but also by local factors, which include spatial conditions, as well as location values related to fashion, security preferences or attractiveness related to a given housing estate or district. Information on increased activity of the real estate market in certain areas may be a signal of the presence of some new factors of local importance (e.g., new subway line [57,65]) influencing the demand (or supply) for real estate.

## 5.3. Mapping Price Dynamics

In the analyzed period, the prices of flats increased on average by 4% annually in Poznań (based on model regression coefficients). Spatial differentiation of local conditions determining prices of apartments may indicate that the dynamics of price changes are not the same in the whole area. Location preferences, the dominant type of development, traffic conditions, and green areas are only examples of many local factors that allow global regression models to make only certain generalization of market trends. The geographically weighted regression model was used to develop a map of spatial differentiation of the dynamics of price changes. During the analysis, it was assumed that the impact of property characteristics, both spatial and non-spatial, was constant throughout the whole area, while the explanatory variable was the time expressed in the number of months that passed from January 2013 to the transaction date. This required the application of adjustments resulting from the QR model, which allowed us to eliminate the impact of other variables. The adjustments were estimated on the basis of average values of explanatory variables. In the model, for evident reasons, the variables indicating the year of the sale were omitted. The GWR model then takes the following form:

$$Y = \beta_0(x_j, y_j) + \sum_{i=1}^{k} \beta_i \cdot (X_i - X_{imean}) + \beta_j(x_j, y_j) \cdot t + \varepsilon_j(x_j, y_j) \tag{10}$$

where $X_{imean}$ denotes the average value of the variable $X_i$, and $\beta_j(x_j, y_j)$ denotes the parameter indicating the monthly trend of price changes determined at the point of coordinates (xj, yj). After taking into account the logarithmic transformation, the average annual price change trend is defined as follows:

$$p = \left[1 - \exp\left(\beta_j\left(x_j, y_j\right)\right)\right] \cdot 12 \tag{11}$$

The use of spatial models, including the GWR model, is justified when the phenomenon of spatial autocorrelation of the explained variable occurs. This means that prices of properties located close to each other should be similar. Figure 5 presents a Moran chart for standardized logarithms of unit transactional prices and the results of the global autocorrelation test (Moran I statistics).

The Moran I test statistic was 0.257. A low p-value means that prices (logarithms from prices) were characterized by significant global spatial autocorrelation. This is an important reason for using spatial models for price analysis.

Figure 6 shows the spatial distribution of the dynamics of price changes expressed as a percentage (left) and the distribution of errors resulting from the GWR model (right).

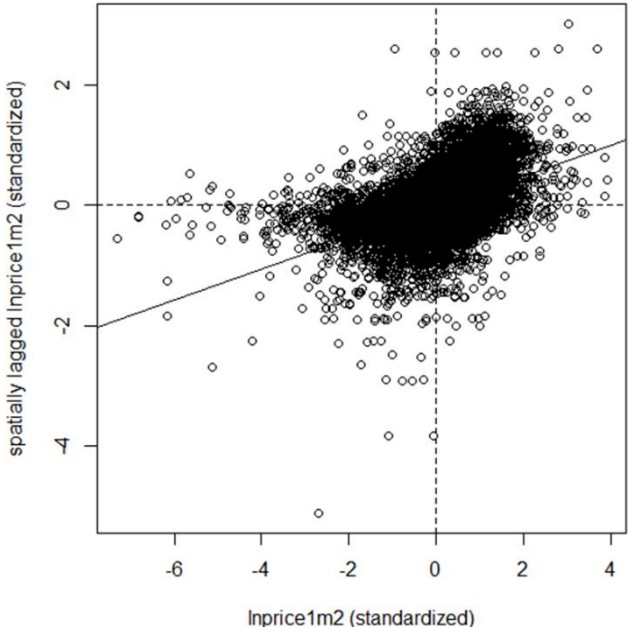

**Figure 5.** Moran's test results for lnprice1m2.

The analysis of the price dynamics map allows for identification of the areas of the city where the dynamics of price changes clearly differ from the typical value (about 4% per year). The map allows identification of areas perceived as more or less attractive in the opinion of buyers. Controlling the physical characteristics and quality elements of the property, the price dynamics maps allow one to highlight and assign price volatility to specific locations. The distribution of errors in determining the price change coefficient is naturally related to the distribution of the number of transactions. The highest error value concerns the areas where only single transactions occurred. It must be stated that Figure 5 (right map) shows the distribution of errors so that it is visible where there are few transactions, and because of that the trend in price changes is more prone to bias. The information value is not only about where the trend is but also about how accurate it is.

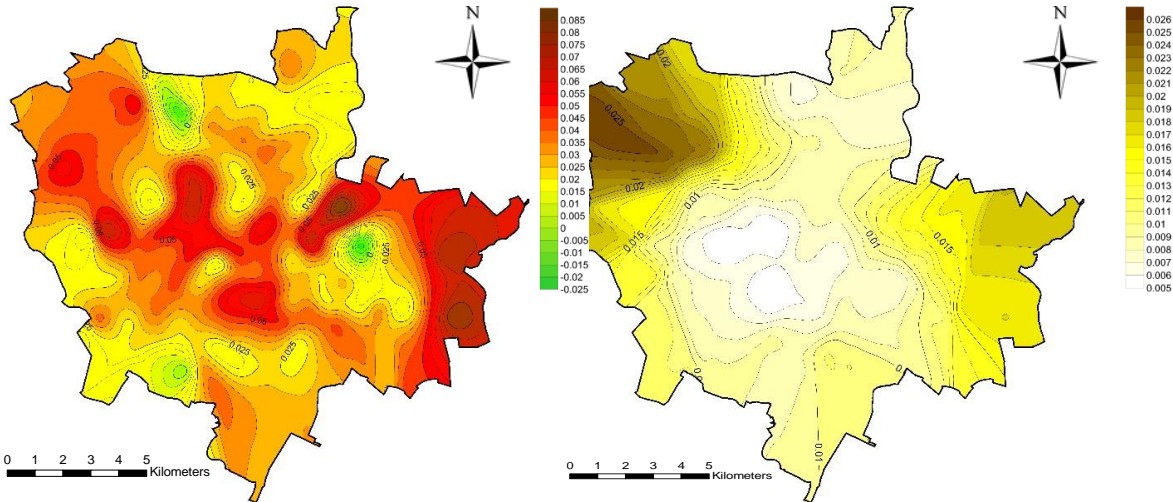

**Figure 6.** Map of the distribution of the average annual price change (1% = 0.01) and the spatial distribution of errors in the price change index.

## 6. Summary and Conclusions

One of the most crucial reasons for imperfections in the real estate market, i.e., its low efficiency and transparency, is insufficient access to information on the current transaction prices and their dynamics, especially in spatial terms. This information can be developed and made available in the form of maps of average prices and market activity. Within the framework of the conducted research and analyses, the concept and principles of cartographic presentation of residential property prices and spatial dynamics of their changes were presented. The use of hedonic models (OLS and QR) made it possible to determine the impact of both non-spatial and spatial factors on prices. The influence, especially on spatial factors, was taken into account when creating the reference price map, which, after taking into account the spatial distribution of the residuals, was the basis for the development of the price map of residential units in the form of an isoline and a cartogram. The map of average prices may at the same time reflect the values of the location determined by the quality of urban infrastructure, distance from specific places, and environmental factors. Market activity maps made with the use of kernel estimation indicate areas where the market is developing dynamically, while information on trends in the number of transactions may indicate the development potential of individual parts of the city, including their growing or decreasing attractiveness. During the study of changes in residential prices resulting from the passage of time, not only the general trend was taken into account, but using the geographically weighted regression model, the spatial distribution of the price index was also presented. A map of existing trends may be one of the principal reasons for making investment decisions in the residential market.

The information on the local housing market presented on the maps may be one of the primary sources of information not only for potential buyers and sellers but also for market professionals, i.e., real estate appraisers, managers, and agents. Of course, the provided information must be up to date and widely available. The presented solutions undoubtedly have the advantages of using relatively simple mathematical instruments and commonly available GIS tools in accordance with the principle of Occam's razor [66], which is sometimes paraphrased by the statement, "the simplest solution is most likely the right one".

In summary, the additional information delivered in residential maps should contribute to easier understanding and interpretation of the phenomena occurring in the real estate market and at the same time should increase the transparency of this market. We are aware of some limitations of the presented procedures; for example, in the case of thin markets with few transactions, the obtained results may be biased.

**Author Contributions:** Conceptualization: Radoslaw Cellmer and Radoslaw Trojanek; Methodology, Radoslaw Cellmer; Investigation, Radoslaw Cellmer and Radoslaw Trojanek; Data Curation, Radoslaw Trojanek; Writing—Original Draft Preparation, Radoslaw Cellmer and Radoslaw Trojanek; Writing—Review and Editing, Radoslaw Cellmer and Radoslaw Trojanek; Visualization, Radoslaw Cellmer. All authors have read and agreed to the published version of the manuscript.

**Funding:** This study is based on a research project financed by the National Science Centre of Poland (grant no. 2017/27/B/HS4/01848, project entitled "Modelling and forecasting of housing prices").

**Conflicts of Interest:** "The authors declare no conflict of interest".

## Appendix A

In this appendix, we present histograms of chosen variables.

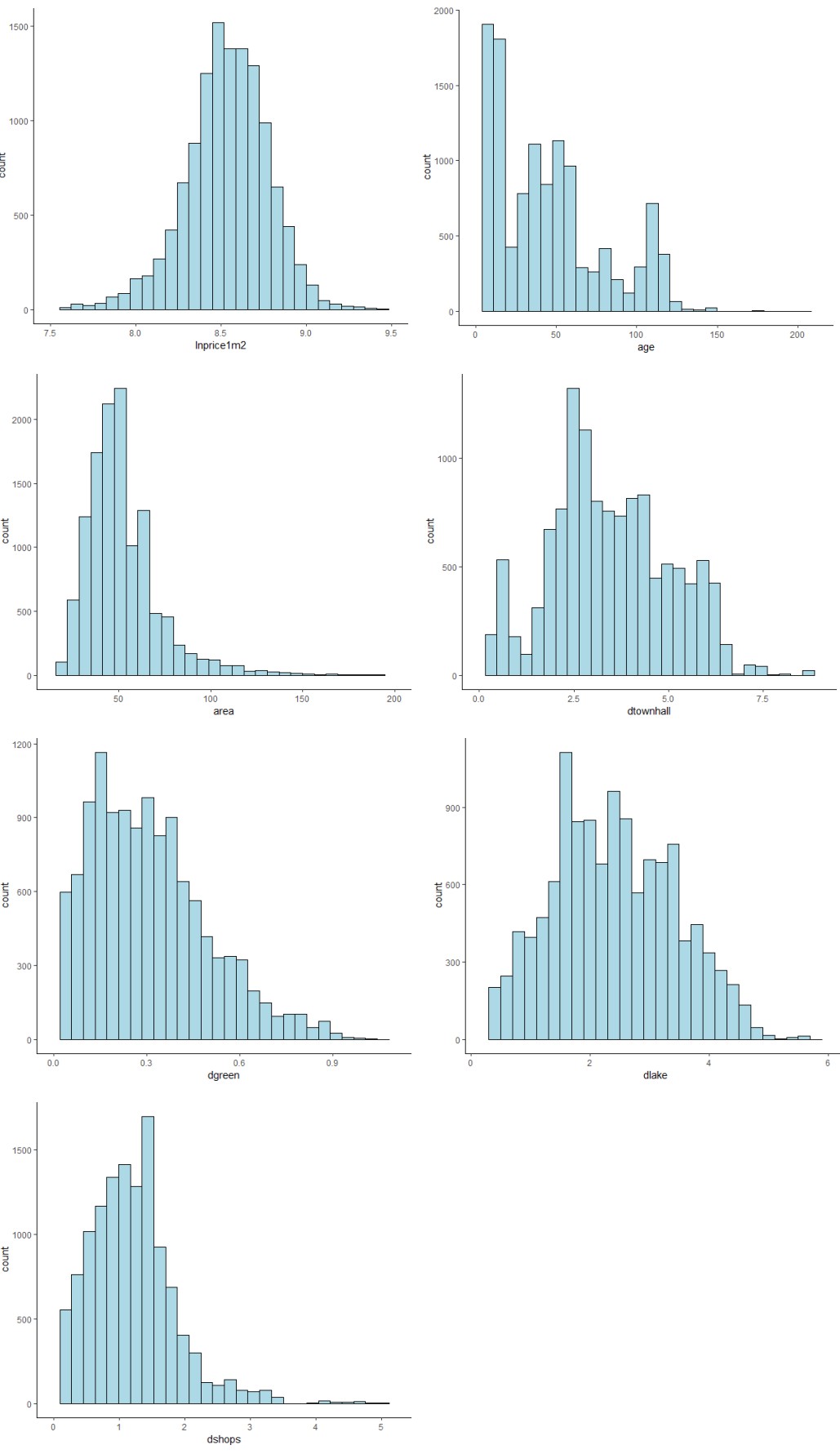

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
