# Peer review of "Towards Increasing Residential Market Transparency: Mapping Local Housing Prices and Dynamics"

_ijgi, doi:10.3390/ijgi9010002_

Round 1
Reviewer 1 Report
Dear colleagues,
This manuscript has a clear aim and a sound research design. The findings are novel and the approach is welcome the scientific community. Below are two lists of comments. The first is the major concerns and the second is minor issues. The major concerns does not necessarily need methodological changes, but should be given appropriate consideration in the reasoning and motivation of the methods used.
One major concern is the use of Kernel densities on highly geographically uneven distributed sales. Since the aim is to help make the property market more transparent by presenting informative maps, it is important to minimize the risk of them being misinterpreted. Superimposing the distribution of sales (Figure 1) over for example the average annual price changes in Figure 5 makes me wonder if the annual changes in some areas are based on a large enough sample of sales to show a transparent image of the real estate market in Poznan. I think the discussion around this is central and should be given more attention in the manuscript. Connected to comment 1, I suggest a motivation of the specification of the smoothing parameter when using Kernel densities. This too is important for the transparency. Perhaps reasoning along the lines of the modifiable areal unit problem. See Unwin (1996), Openshaw (1984) and Fotheringham and Wong (1991) The sample consists of apartment sales, which is only a part of the housing market. This must be acknowledged more in the reasoning of the significance of this study’s results. My interpretation is that the study shows a proof of concept of how to create informative maps of the residential market, but using only apartments. The exclusion of single-family houses must be explicitly mentioned and the conclusions from the study have to be interpreted as such. Using data from Open Street Map is often fine if the availability is ok, but the risks of using geographic data from volunteers should be mentioned. The relationship between housing prices and distance to amenities are most commonly non-linear, which should be acknowledged in the paper. The quality of amenities should also be recognized an an important factor. For example the already cited paper by Czembrowski and Kronenberg (2016) makes the case that the size of the parks has a significant effect on housing prices. The description of how the data was collected, for example the use of street view, should be clearer. For reference on the use of street view in hedonic modelling see Law, S., Paige, B., & Russell, C. (2018).
Minor fixes:
On line 246-249 there is a repetition. Add scale or scale bar and north arrow on all maps. The abstract refers to “spatial maps”. Perhaps geographic maps or just maps is more appropriate? Repetition on line 118. The description of the weight matrix on line 230-233 should be clearer.

Reviewer 2 Report
MAJOR COMMENTS
To what extent could your analysis contribute to increasing residential market transparency? I can see that you estimated the price or residential units, but I do not see the connection between them.
You did not provide any research statement, findings and discussion in the abstract.
I am wondering how density mapping (kernel density) could help to improve transparency?
You did not summarize your findings in the summary and conclusions.
You may wish to re-organize the method and results sections. You have provided equations in results sections that should be in methods. For example, lines 289- 309 are about methods.
Although you did OLS and GWR, you did not provide why GWR need to be performed. You need to show autocorrelation of housing price first.
Please provide the data source in Data section.
It would be better to show the patterns of variables through histogram (Table 2).
MINOR COMMENTS
“The research included the development of a map of real estate prices, a map of market activity and a map of the dynamics of price changes. R environment and ArcGIS v. 10.4.1 software from ESRI were used for analyses and visualization” should be in Method section (line 269-271).
There is no scale in Figure 2, 3, and 4.
Reviewer 3 Report
This research aimed to make three types of spatial maps of local housing prices to increase residential market transparency. The maps included a map of housing prices, a map of market activity, and a map of the dynamics of price changes. As a method, the quantile regression model was used for the first map, kernel density estimation was applied for the second map, and geographically weighted regression was used for the third map. Transaction records in Poznan, Poland between 2013 and 2017 were used.
Overall, the results of visualizing the spatial housing price on the map were somewhat useful. However, the manuscript will need to be edited for publication.
It may be better to clarify the motivation of the research. Above all, no clear reason why the three maps were created was given in the introduction. The purpose such as "to increase information transparency" is not specific. In this sense, the novelty of visualizing three maps should be emphasized somewhere. According to literature review in this study, there exist related studies using each method, so it is hard to know what novelty this study brings.
Regarding Introduction, the sections before "in recent years (line 60)" may need to be rearranged. The present tense was used, but was it in the present? It should be in the past. It was also difficult to determine whether the described problem was a global problem or a local one. In addition, it is hard to understand what the imperfection of the real estate market (line 41) or the lack of transparent market information (line 49-50) specifically means. In Results, the introduction of the software used and equations 8-12 should be moved. Many parts have to go to the method.
Reviewer 4 Report
The article presented for review is an interesting case study. In my opinion, the article may be published after minor amendments have been made:
1) Please provide all prices in a more recognizable currency (EUR or USD) or provide information about the current exchange rate of PLN.
2) In the whole article: in "m2" - "2" in the top index.
Moreover, maybe in the "Literature review" section it is worth noting the issues described in:
- http://yadda.icm.edu.pl/yadda/element/bwmeta1.element.desklight-8578bc58-185a-474d-9c3d-078f7eb99840
- http://yadda.icm.edu.pl/yadda/element/bwmeta1.element.baztech-7cafd953-d2a8-435e-9ea1-cd23b395e2c3
- http://yadda.icm.edu.pl/yadda/element/bwmeta1.element.ekon-element-000171307057
Author Response
Please see the attachment.

This manuscript is a resubmission of an earlier submission. The following is a list of the peer review reports and author responses from that submission.